# The Probiotic Strain *Clostridium butyricum* TO-A Produces Butyrate by Utilizing Lactate and Acetate

**DOI:** 10.3390/ijms26072951

**Published:** 2025-03-24

**Authors:** Shotaro Honda, Hiromichi Eguchi, Yoichi Okino, Dian-Sheng Wang

**Affiliations:** Research Division, TOA Biopharma Co., Ltd., 606 Kondoh-cho, Tatebayashi 374-0042, Japanokino@toabio.co.jp (Y.O.); dwang@toabio.co.jp (D.-S.W.)

**Keywords:** *Clostridium butyricum* TO-A, probiotics, lactate-utilizing bacteria, lactate-utilizing, butyrate, lactate, acetate, racemization

## Abstract

Lactate-utilizing bacteria (LUB) are intestinal bacteria that produce butyrate from lactate and acetate, key metabolites in the gut. As LUB help maintain lactate and butyrate concentrations in the intestinal tract, they are promising probiotic candidates. *Clostridium butyricum* TO-A (CBTOA) has reportedly been effective in treating various gastrointestinal issues in humans and animals. Although CBTOA is known to increase intestinal butyrate levels, it is unclear how it utilizes lactate and acetate, similar to LUB, to produce butyrate. We investigated lactate utilization-related genes in CBTOA and examined the relationship between lactate and acetate utilization and butyrate production using peptone–yeast medium supplemented with D-lactate, L-lactate, and/or acetate. This study demonstrates for the first time that the probiotic strain CBTOA harbors lactate utilization-related genes and efficiently produces butyrate only in the presence of exogenous lactate and acetate instead of sugars. Furthermore, CBTOA expresses a lactate racemase that enables the bacterium to utilize both lactate enantiomers while regulating the ratio of D-lactate to L-lactate in the intestinal microenvironment via racemization. In conclusion, CBTOA efficiently produces butyrate utilizing lactate and acetate, similar to LUB; therefore, CBTOA could be an efficient butyrate supplier as a probiotic strain in the intestinal tract.

## 1. Introduction

Butyrate, a short-chain fatty acid (SCFA), serves as a source of energy for colonocytes and contributes to colon homeostasis [1,2,3,4]. Butyrate plays a number of important roles in the intestines, including enhancing the barrier function of intestinal epithelial cells [5,6], promoting colonic mucin secretion [6], stimulating the production of antimicrobial peptides [7], and alleviating colitis [4,8,9]. In the colon, butyrate is produced by intestinal bacteria, and dysbiosis results in decreased levels of butyrate in the feces [2,10]. Maintaining an appropriate butyrate concentration in the intestinal tract thus requires the normal function of intestinal bacteria.

Intestinal bacteria produce SCFAs such as butyrate and acetate primarily by utilizing dietary fiber [2]. However, several recent studies reported that in addition to dietary fiber, some intestinal bacteria can produce butyrate from lactate and acetate, which are fermentation products of intestinal bacteria [11,12]. These bacteria are defined as lactate-utilizing bacteria (LUB). Despite the presence of lactic acid bacteria within the gut microbiota, only low levels of lactate are normally detected in the feces [11]. LUB are thought to play a role in maintaining appropriate concentrations of lactate and butyrate in the intestinal tract and thus have attracted considerable attention as candidate probiotic strains [13,14]. Weghoff et al. reported that the lactate dehydrogenase (LDH) of *Acetobacterium woodii* DSM 1030 converts lactate to pyruvate in the presence of electron-transfer flavoprotein (Etf) A, Etf B, and reduced ferredoxin [15]. After that, several studies have proposed a pathway for lactate utilization by LUB [11,15], in which extracellular lactate is taken up by LUB and converted to pyruvate via the abovementioned reaction. Pyruvate is then metabolized to butyryl-CoA via the conversion of acetyl-CoA in the butyrate metabolic pathway. Finally, butyryl-CoA is converted to butyrate via the butyrate kinase or butyryl-CoA–acetate-CoA transferase pathway. Similar lactate-utilizing pathways have been proposed for other butyrate-producing bacteria [13,16,17]. Additionally, the genes encoding LDH and Etf are conserved, along with butyrate production-related genes, and LUB reportedly harbor this gene cluster [13,15,16,17].

*Clostridium butyricum* is an obligate anaerobic, Gram-positive bacterium that particularly can produce butyric acid among the SCFAs [18,19,20]. Additionally, *C. butyricum* forms spores, exhibiting excellent acid resistance [19,20], which is thought to enable *C. butyricum* to reach the colon alive. Furthermore, other studies have reported that butyrate produced by *C. butyricum* plays a role in suppressing the expression of inflammatory cytokines during the onset of enteritis via regulation of TLR4 expression in intestinal epithelial cells [18,20]. Due to these characteristics, *C. butyricum* has garnered significant attention as a potential probiotic species, and formulations containing this single strain or combinations with other probiotic strains have been used clinically for over 60 years.

*Clostridium butyricum* TO-A (CBTOA) is commonly employed as a probiotic strain. Probiotic formulations containing CBTOA are reportedly effective in treating diarrhea, constipation, enteritis, and other intestinal diseases or symptoms in both humans and animals [21,22,23,24,25,26,27]. A recent study has also shown that CBTOA culture supernatant downregulated the expression of several inflammation- and DNA replication/repair-related genes in human cervical cancer HeLa S3 cells [28]. Moreover, as CBTOA lacks *Clostridium botulinum* neurotoxin genes, probiotics containing this strain are safe and highly efficacious in pediatric patients with infectious gastroenteritis [26,29]. Inatomi et al. reported that the administration of CBTOA-containing probiotics to Japanese Black cattle increased the amount of butyrate in the feces, indicating an increased concentration of butyrate in the intestines [30]. However, despite the contribution of CBTOA to increased intestinal concentrations of butyrate, whether CBTOA utilizes lactate and acetate to produce butyrate, similar to LUB, remains unclear.

The purpose of this study was to determine whether the probiotic strain CBTOA utilizes lactate and acetate to produce butyrate in a manner similar to LUB. Therefore, CBTOA was cultured in a medium supplemented with lactate and/or acetate instead of sugars and the butyrate production was then assessed. In addition, lactate-producing bacteria can produce either D-lactate, L-lactate, or both. We hypothesized that these lactate enantiomers can affect lactate utilization by CBTOA and therefore evaluated the effect of D- and L-lactate on lactate utilization and butyrate production by CBTOA.

## 2. Results

### 2.1. Sequence Search and Comparison of Putative Lactate Utilization-Related Genes of CBTOA

We searched the genome of CBTOA for homologs with lactate utilization-related genes of *Clostridium butyricum* KNU-L09 (CBKNU-L09) [17] and found that the genes encoding lactate racemase, Etf B, Etf A, lactate permease, D-LDH, acryl-CoA dehydrogenase, and others were conserved in the genome of CBTOA (Figure 1). The proteins translated from these putative lactate utilization-related genes of CBTOA exhibited 100% amino acid identity with the lactate utilization-related proteins expressed in CBKNU-L09. In addition, the amino acid sequences of the lactate utilization-related proteins of CBTOA were compared with those of *Acetobacterium woodii* DSM 1030, *Anaerostipes hadrus* SSC/2, and *Anaerobutyricum hallii* and exhibited > 30% identity with D-LDH, Eft A, and Eft B, which are involved in the reverse reaction in each strain [11,13,15].

### 2.2. Evaluation of Lactate Utilization by CBTOA

To evaluate lactate utilization by CBTOA, cells were cultured in a medium supplemented with lactate and acetate instead of sugars. CBTOA were capable of growth in peptone–yeast (PY) medium and PY medium supplemented with acetate, D-lactate, and L-lactate (PY + ADL) (Figure 2, Table 1). Furthermore, an increase in pH was observed beginning at 6 h after cultivation initiation in the PY + ADL medium. The pH of the PY medium after 16 h of cultivation was 5.94 ± 0.01, whereas the pH of the PY + ADL medium was 6.24 ± 0.08. Although the OD_600_ of the PY medium and PY + ADL medium at the end of cultivation was almost the same, the total viable count of CBTOA in the PY + ADL medium was approximately twice that in the PY medium (Table 1). By contrast, the sporulation rate was similar in both media, at approximately 3% (Table 1).

Regarding the production of organic acids, the concentrations of lactate, acetate, and butyrate after 16 h of cultivation in the PY medium were higher than the concentrations at the start of cultivation (Table 2). In the case of the PY + ADL medium, acetate and butyrate concentrations increased similarly to the PY medium, but the lactate concentration decreased. The concentration of butyrate produced by CBTOA in the PY + ADL medium was 4.94 ± 0.83 mM, which was significantly higher than the 2.59 ± 0.10 mM produced in the PY medium (*p* < 0.008).

### 2.3. Effect of Lactate Enantiomers on Lactate Utilization by CBTOA

To investigate the effect of lactate enantiomers on lactate utilization by CBTOA, we cultured CBTOA in a PY medium supplemented with acetic acid and D-lactate (PY + AD) or acetic acid and L-lactate (PY + AL). CBTOA grew in both PY + AD and PY + AL medium (Table 3, Figure 3). Furthermore, the pH of the PY + AD medium increased during cultivation, similar to the PY + ADL medium (Figure 2). However, no such pH increase was observed in the PY + AL medium (Figure 3). After 16 h of cultivation, the pH of the PY + AD medium was 6.43 ± 0.03, whereas the pH of the PY + AL medium was 6.04 ± 0.08. Although both media exhibited similar OD_600_, the total viable count was higher in the PY + AD medium than in the PY + AL medium (Table 3). By contrast, the sporulation rate of CBTOA was higher in the PY + AL medium than in the PY + AD medium (Table 3). After 16 h of cultivation in the PY + AD medium, the butyrate concentration was significantly higher compared with the start of cultivation; however, no change in acetate concentration was observed. The concentration of lactate was lower after 16 h compared with the start of cultivation in the PY + AD medium (Table 4). In the PY + AL medium, by contrast, CBTOA produced only a small amount of lactate and significantly more acetate and butyrate (Table 4). Furthermore, butyrate production by CBTOA was 7.54 ± 0.67 mM in the PY + AD medium and 2.96 ± 0.10 mM in the PY + AL medium. A comparison of the net production of organic acids (lactate, acetate, and butyrate) in each medium revealed that CBTOA exhibited increased butyrate production in the presence of lactate and acetate (Figure 4). In addition, butyrate production by CBTOA increased depending on the supplementation of the medium with D-lactate, whereas lactate and acetate production decreased. However, no difference in net butyrate production was observed between the PY medium (2.59 ± 0.10 mM) (Table 2) and the PY medium supplemented with acetate alone (PY + A, 2.79 ± 0.02 mM) (Appendix A) or the PY medium supplemented with D-lactate alone (PY + D, 2.91 ± 0.23 mM) (Appendix A).

### 2.4. CBTOA Lactate Racemic Reaction

Because lactate enantiomers in the culture medium affect the butyrate production of CBTOA, we investigated whether the composition of lactate enantiomers changes after CBTOA cultivation by measuring the concentrations of D-lactate and L-lactate and determining their ratios in various media. CBTOA was cultured in PY, PY + AD, PY + AL, or PY + ADL media for 16 h, and then the concentrations of L-lactate and D-lactate were measured using an F-kit, as described in Section 4. Compared with the start of cultivation, the composition of lactate enantiomers changed in all media except PY + ADL. The ratio of lactate enantiomers (L-lactate to D-lactate, with a total of 100) in the PY medium changed from 70:30 at the start of cultivation to 53:47 after cultivation, from 4:96 to 41:59 in the PY + AD medium, and from 99:1 to 70:30 in the PY + AL medium (Figure 5). No pre-versus post-cultivation change in the lactate enantiomer ratio was observed in negative control media incubated without CBTOA (Appendix A).

## 3. Discussion

Although CBTOA has been used as a probiotic strain and products containing it are reportedly useful according to several clinical studies, little research has focused on characterizing the ability of CBTOA to utilize lactate and acetate, which are critical organic acids for butyric acid production in the colon [23,25,26,27]. In this study, we report for the first time that CBTOA not only utilizes sugars but also lactate and acetate to produce butyrate. The genome of CBTOA contains a putative lactate utilization-related gene cluster exhibiting 100% identity with that of CBKNU-L09 [17]. Consistent with this genomic feature (Figure 1), CBTOA increased butyrate production in the PY medium containing both lactate and acetate (PY + ADL, PY + AD, or PY + AL) compared to PY medium alone (Table 2 and Table 4), whereas butyrate production was only slightly increased in the PY medium containing either lactate or acetate alone (PY + D or PY + A) (Appendix A). Although CBTOA is capable of producing acetate (1.63 mM) by itself in PY medium (Table 2), the net acetate production (0.13 mM) was decreased when both D-lactate and acetate were supplemented in the PY medium (PY + AD) (Table 4), suggesting that exogenously supplied acetate was consumed along with lactate. These findings indicate that CBTOA requires the co-utilization of both the lactate and acetate of substrates for effective butyrate production. Notably, a previous study reported that CBKNU-L09 also depends on exogenous lactate and acetate to produce butyrate [17], and the simultaneous requirement of both substrates is consistent with our observations. Thus, our results reveal a previously uncharacterized function of CBTOA, reinforcing the importance of lactate and acetate as simultaneous substrates for butyric acid production.

The pH of all media examined in this study decreased during the early stages of CBTOA cultivation (Figure 2 and Figure 3). In the later stages of cultivation, the pH of the PY + AD medium and PY + ADL medium increased, whereas the pH of the PY medium and PY + AL medium did not change. It has been reported that lactate utilization is associated with an increase in pH [31,32], which is consistent with the observation that pH increased only in the PY + AD and PY + ADL media, in which lactate appeared to be consumed in this study. Moreover, total viable counts of the PY + ADL (Table 1) and PY + AD (Table 3) media were higher than those in the PY (Table 1) and PY + AL (Table 3) media, suggesting that the pH rise was not due to cell death but rather due to the metabolic conversion of lactate into less acidic end products. These results thus suggest that utilization of lactate by CBTOA increases the pH of the medium.

The net lactate production by CBTOA in the PY, PY + AL, PY + ADL, and PY + AD media was 1.14, 0.36, −2.42, and −5.57 mM, respectively, and the net butyrate production was 2.59, 2.96, 4.94, and 7.54 mM, respectively (Table 2 and Table 4). Furthermore, the presence of D-lactate in the medium affected the lactate utilization and butyrate production of CBTOA (Figure 4), and a putative CBTOA lactate utilization-related gene cluster was shown to contain genes encoding D-LDH (Figure 1). Several previous studies proposed that D-LDH converts D-lactate to pyruvate [13,16,17]. The results of this study, which showed increased lactate consumption and butyrate production by CBTOA in the presence of D-lactate, are consistent with those of previously reported studies. Therefore, the preferential utilization of D-lactate by CBTOA is a very valuable characteristic for its use as a probiotic strain. D-lactate accumulates in the intestinal tract of patients with short bowel syndrome and in the rumen of ruminants, causing lactic acidosis [13,33,34]. Particularly in dairy cows, rumen acidosis reduces milk production, leading to economic losses. Thus, CBTOA, which preferentially utilizes D-lactate, could be expected to serve as a consumer of D-lactate in vivo.

In the medium cultured with CBTOA, lactate was racemized, and the ratio of lactate enantiomers was regulated to maintain balance (Figure 5). By contrast, racemization did not occur in media without CBTOA (Appendix A). These results suggest that CBTOA has the ability to racemize lactate. A previous study reported that *A. hadrus* SSC/2, which does not harbor a gene encoding lactate racemase in its genome, cannot utilize L-lactate, and LUB are thought to require lactate racemase to utilize L-lactate [13]. In the CBTOA genome search conducted in the present study, a putative lactate racemase gene was identified upstream of a cluster of putative lactate utilization-related genes (Figure 1). The putative lactate racemase exhibited 100%, 57%, and 53% identity with the lactate racemases of CBKNU-L09, *A. woodii* DSM1030, and *A. hallii*, respectively. The CBTOA lactate racemase also showed 58% identity with the lactate racemase of *Lactobacillus plantarum* (PDB no. 5HUQ) [35], which has been registered in the Protein Data Bank (https://www.rcsb.org/, accessed on 14 April 2021) with details regarding its activity and crystal structure. Additionally, in the PY medium, the net lactate and net butyrate production by CBTOA were 1.14 and 2.59 mM, respectively, higher than the net lactate production of 0.36 mM but lower than the net butyrate production of 2.96 mM in the PY + AL medium. These results suggest that CBTOA converts L-lactate to D-lactate using the putative lactate racemase, thereby producing butyrate.

This study showed that CBTOA possesses a lactate-utilization pathway similar to that found in LUB and efficiently produces butyrate by utilizing lactate and acetate instead of sugars. A previous study reported that the direct delivery of butyrate to the colon has positive effects in the treatment of inflammatory bowel disease [36]. Therefore, the use of LUB has been proposed as a new probiotic therapeutic approach for treating inflammatory bowel disease [14]. Interestingly, the use of probiotics containing CBTOA was shown to improve irritable bowel syndrome-like symptoms in patients with ulcerative colitis in endoscopic remission [27]. However, it is important to note that the study lacked a placebo control group and did not test CBTOA alone. Thus, CBTOA, which can utilize lactate, could be a potential supplier of butyrate to the colon as a probiotic strain. However, in the feces of patients with severe ulcerative colitis, lactate reportedly accumulates, while the concentrations of organic acids such as acetate decrease [37]. The necessity of both lactate and acetate for lactate utilization by CBTOA suggests that the ratio of lactate to acetate in the intestinal microenvironment may limit the lactate utilization of this organism.

## 4. Materials and Methods

### 4.1. Sequence Search and Comparison of Putative Lactate Utilization-Related Genes in CBTOA

Based on the sequence of the lactate utilization-related genes of *C. butyricum* KNU-L09 (CBKNU-L09) (17), BLAST (https://blast.ncbi.nlm.nih.gov/Blast.cgi, Version: BLAST+ 2.13.0) was used to search for putative lactate utilization-related genes using the sequences of the two chromosomes of CBTOA (accession no. CP014704, CP014705) stored in Genbank (https://www.ncbi.nlm.nih.gov/genbank/, accessed on 20 January 2020). The amino acid sequences of CBTOA putative lactate utilization-related proteins were compared with those of CBKNU-L09, *Acetobacterium woodii* DSM 1030, *Anaerostipes hadrus* SSC/2, and *Anaerobutyricum hallii* using NCBI Protein BLAST, and the Identity with the lactate utilization-related genes of CBTOA was then calculated [11,13,15]. The detailed sequences of the lactate utilization-related genes used for comparison are shown in Appendix A.

### 4.2. Materials

#### 4.2.1. Bacteria Strain

This study used the probiotic strain CBTOA (TOA Biopharma Co., Ltd., Tokyo, Japan) [18,21,22,23,24,25,26]. CBTOA was stored in the spore state at −80 °C in a preservation solution containing 10% dimethyl sulfoxide.

#### 4.2.2. Basic Medium

Peptone–yeast (PY) medium, which is PYG medium without glucose [18], was used as the base medium. The PY medium consisted of 1% yeast extract (Becton, Dickinson and Company, BD, Franklin Lakes, NJ, USA), 0.5% peptone (BD), and 0.5% tryptone (BD).

### 4.3. Growth Conditions for CBTOA

CBTOA spores (100 µL) were inoculated into a sterile PY medium and heat shocked at 80 °C for 10 min. The spores were then pre-cultured in a TE-HER Anaerobox (Hirasawa, Tokyo, Japan) under anaerobic conditions at 37 °C for 4 h. For the culture of CBTOA, a multi-channel fermenter (Bio Jr. 8, ABLE, Tokyo, Japan) was used. A total of 80 mL of the medium was autoclaved in a Bio Jr. 8 vessel at 121 °C for 20 min. To maintain anaerobic conditions until the start of cultivation, sterilized N_2_ gas was blown into the Bio Jr. 8 vessel through a 0.25-µm PTFE filter (Membrane Solutions, Plano, TX, USA).

Cultures were initiated by inoculating 800 µL of CBTOA pre-cultured suspension into the medium and incubating it at 37 °C and 150 rpm with continual pH measurement. After 16 h of cultivation, the bacterial suspension was sampled to measure the OD_600_, total viable count, and spore count. A portion of the culture suspension was centrifuged (6000× *g*, 5 min), and the concentrations of organic acids and lactate enantiomers in the supernatant were measured. All cultures in this study were performed in triplicate in independent experiments.

### 4.4. Modified Medium

Depending on the purpose of the experiment, D-lactic acid (D, JMTC, Tokyo, Japan), L-lactic acid (L, Wako, Tokyo, Japan), and/or sodium acetate 3H_2_O (A, Wako, Tokyo, Japan) were added to the PY medium. All media used in this study were adjusted to a pH of 6.85−7.15 using NaOH.

### 4.5. Evaluation of Lactate Utilization by CBTOA

To evaluate lactate utilization, CBTOA cells were cultured under anaerobic conditions at 37 °C and 150 rpm in PY medium or PY + ADL medium, which is PY medium supplemented with 0.25% D-lactic acid, 0.25% L-lactic acid, and 0.5% sodium acetate 3H_2_O.

### 4.6. Effect of Lactate Enantiomers on Lactate Utilization by CBTOA

The effect of lactate enantiomers on lactate utilization by CBTOA was evaluated by culturing CBTOA under anaerobic conditions at 37 °C and 150 rpm in PY + AL medium, which is PY medium supplemented with 0.5% L-lactic acid and 0.5% sodium acetate 3H_2_O, or PY + AD medium, which is PY medium supplemented with 0.5% D-lactic acid and 0.5% sodium acetate 3H_2_O.

### 4.7. Measurement of Total Viable Count and Spore Count

To measure the total viable count, CBTOA culture suspension was serially diluted with phosphate-buffered saline (−) and spread on BL agar medium with 5% horse blood. The plates were then incubated under anaerobic conditions at 37 °C for 24 h in a TE-HER Anaerobox (Hirasawa). To determine the spore count, serially diluted samples were heat-treated at 80 °C for 10 min and then subjected to the same procedure used for total viable count determination. The sporulation rate was calculated as the ratio of the spore count to the total viable count of CBTOA, as follows: sporulation rate (%) = spore count/total viable count × 100.

### 4.8. Measurement of Organic Acid Concentration in Culture Supernatant

The concentrations of various organic acids (lactate, acetate, and butyrate) in the culture supernatant were measured using an HPLC system (Prominence, Shimadzu, Kyoto, Japan) equipped with an electrical conductivity detector (CDD-10A, Shimadzu). The HPLC setup included two ion-exclusion columns (Shim-pack Fast-OA; 7.8 mm × 100 mm, Shimadzu) and a guard column (Shim-pack Fast-OA[G]; 4.0 mm × 10 mm, Shimadzu). The mobile phase and pH buffer were 5 mM *p*-toluenesulfonic acid (Shimadzu) and 5 mM *p*-toluenesulfonic acid, 20 mM bis-Tris, and 0.1 mM EDTA, (Shimadzu), respectively. The flow rate was set at 0.8 mL min^−1^, and the column temperature was maintained at 50 °C.

### 4.9. Measurement of D-Lactate and L-Lactate Concentrations in Culture Supernatant

The concentration of D-lactate or L-lactate in the culture supernatant was measured using an F-kit (J.K. International, Tokyo, Japan) according to the manufacturer’s instructions.

### 4.10. Statistical Analysis

Results are shown as the mean ± standard deviation (SD) of three independent experiments. The Student’s *t*-test was performed using Excel software 2016 (Microsoft Corp., Redmond, WA, USA).

## 5. Conclusions

The present study showed for the first time that CBTOA, which is currently used as a probiotic strain, efficiently produces butyrate in the simultaneous presence of exogenous lactate and acetate instead of sugars. Furthermore, these results suggest that CBTOA, which expresses a lactate racemase, can utilize both lactate enantiomers while regulating the ratio of D-lactate to L-lactate in the intestinal microenvironment via the racemization reaction. Thus, CBTOA could be an efficient butyrate supplier as a probiotic strain in the intestinal tract.

## Figures and Tables

**Figure 1 ijms-26-02951-f001:**
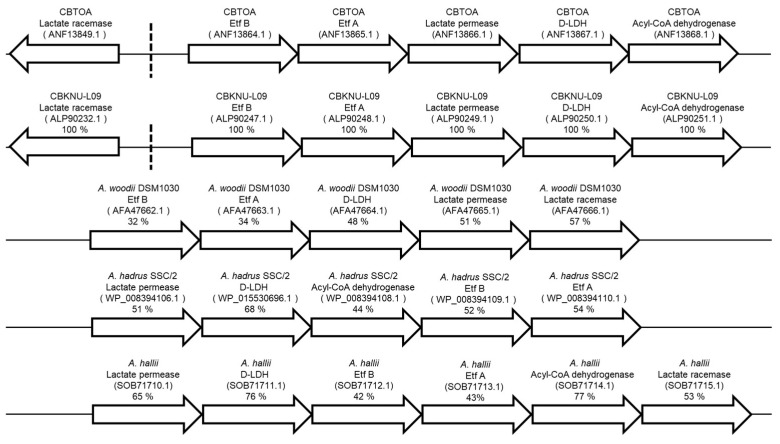
Composition of putative lactate utilization-related gene loci in CBTOA and bacteria already reported to utilize lactate. Dashed lines indicate that the lactate racemase gene of CBTOA or CBKNU-L09 is located in a different region. Arrows indicate the direction of gene translation. Right-facing arrows represent the forward strand, while left-facing arrows represent the reverse strand. Etf A: electron transfer flavoprotein A; Etf B: electron transfer flavoprotein B; D-LDH: D-lactate dehydrogenase; %: indicates percent identity with the lactate utilization-related proteins of CBTOA.

**Figure 2 ijms-26-02951-f002:**
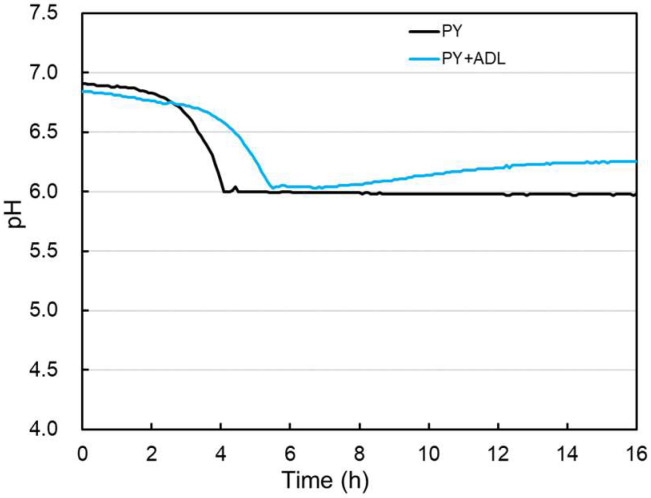
Change in pH of PY medium and PY + ADL medium used to culture CBTOA.

**Figure 3 ijms-26-02951-f003:**
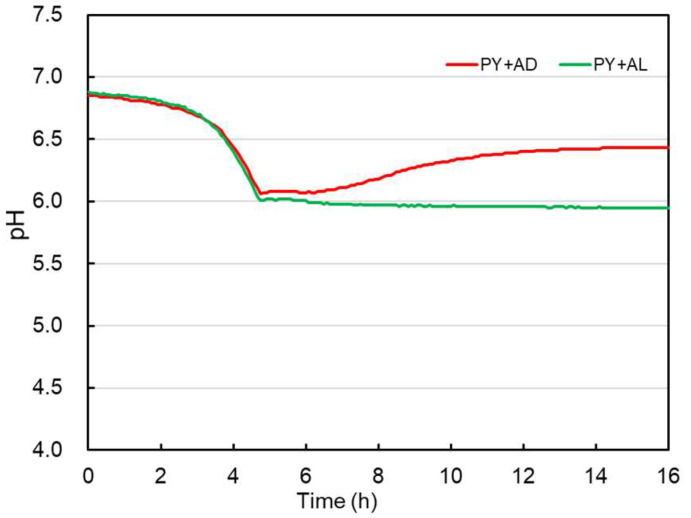
Change in pH of CBTOA cultures in PY + AD medium and PY + AL medium.

**Figure 4 ijms-26-02951-f004:**
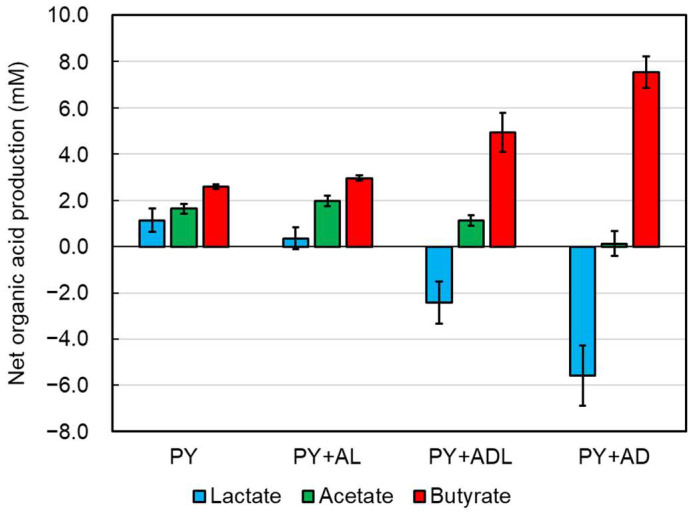
Comparison of net organic acid production by CBTOA in various media. The figure shows the net organic acid production by CBTOA in each culture medium. Blue bars—lactate; green bars—acetate; red bars—butyrate. Error bars indicate SD.

**Figure 5 ijms-26-02951-f005:**
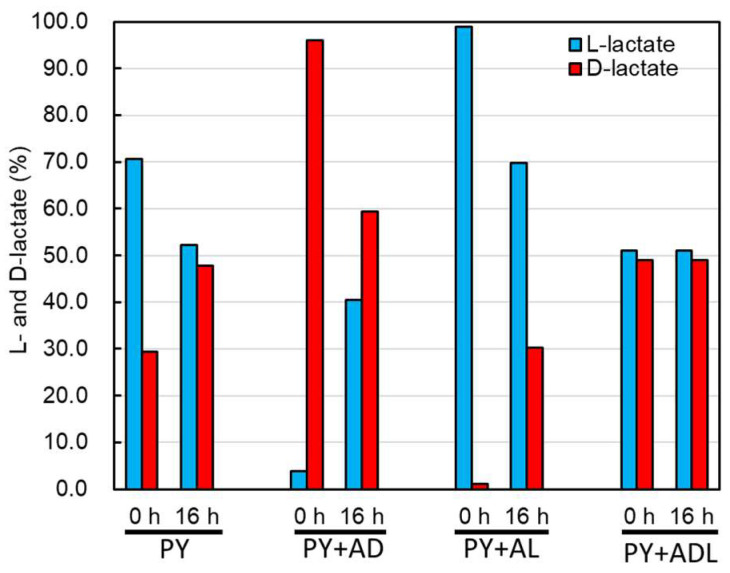
Ratios of lactate enantiomers in various media at the start (0 h) and after 16 h (16 h) of the cultivation of CBTOA. The percentages of each lactate enantiomer in the culture medium are shown. Blue bars—L-lactate; red bars—D-lactate.

**Table 1 ijms-26-02951-t001:** CBTOA-related parameters in PY medium or PY + ADL medium after 16 h of cultivation.

Medium	pH	OD_600_	Total Viable Count (CFU mL^−1^)	Spore Count (CFU mL^−1^)	Sporulation Rate (%)
PY	5.94 ± 0.05	0.57 ± 0.01	7.69 × 10^6^	2.41 × 10^5^	3.13
PY + ADL	6.24 ± 0.08	0.48 ± 0.05	1.32 × 10^7^	4.36 × 10^5^	3.31

*n* = 3. Data are presented as mean ± SD.

**Table 2 ijms-26-02951-t002:** Organic acid production by CBTOA in PY medium and PY + ADL medium. Concentrations and net production of organic acids in PY medium and PY + ADL medium at the start of cultivation (0 h) and after 16 h of cultivation (16 h). *n* = 3. Data are presented as mean ± SD.

Medium	Organic Acid	Concentration (mM)		*p*-Value	Net Production (mM)
		0 h	16 h		
PY	Lactate	2.78 ± 0.52	3.92 ± 0.08	0.059	1.14 ± 0.50
	Acetate	0.94 ± 0.16	2.57 ± 0.08	0.006	1.63 ± 0.22
	Butyrate	0.10 ± 0.01	2.69 ± 0.10	0.0004	2.59 ± 0.10
PY + ADL	Lactate	43.96 ± 0.62	41.55 ± 0.76	0.044	−2.42 ± 0.91
	Acetate	34.60 ± 0.61	35.73 ± 0.39	0.012	1.12 ± 0.22
	Butyrate	0.11 ± 0.02	5.05 ± 0.82	0.009	4.94 ± 0.83

**Table 3 ijms-26-02951-t003:** CBTOA-related parameters in PY + AD medium or PY + AL medium after 16 h of cultivation.

Medium	pH	OD_600_	Total Viable Count (CFU mL^−1^)	Spore Count (CFU mL^−1^)	Sporulation Rate (%)
PY + AD	6.43 ± 0.03	0.56 ± 0.02	1.72 × 10^7^	9.34 × 10^5^	5.44
PY + AL	6.04 ± 0.08	0.43 ± 0.04	4.37 × 10^6^	2.68 × 10^5^	6.13

*n* = 3. Data are presented as mean ± SD.

**Table 4 ijms-26-02951-t004:** Organic acid production by CBTOA in PY + AD medium and PY + AL medium. Concentration and net production of organic acids in PY + AD medium and PY + AL medium at the start of cultivation (0 h) and after 16 h of cultivation (16 h). *n* = 3. Data are presented as mean ± SD.

Medium	Organic Acid	Concentration (mM)		*p*-Value	Net Production (mM)
		0 h	16 h		
PY + AD	Lactate	44.50 ± 0.95	38.93 ± 0.37	0.018	−5.57 ± 1.31
	Acetate	34.22 ± 0.48	34.35 ± 0.07	0.719	0.13 ± 0.54
	Butyrate	0.10 ± 0.00	7.64 ± 0.67	0.003	7.54 ± 0.67
PY + AL	Lactate	44.05 ± 1.29	44.40 ± 1.38	0.327	0.36 ± 0.48
	Acetate	34.82 ± 1.47	36.80 ± 1.25	0.004	1.97 ± 0.23
	Butyrate	0.11 ± 0.01	3.07 ± 0.10	0.0004	2.96 ± 0.10

## Data Availability

Data are contained within the article and Appendix A.

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
