# Peer review of "The Probiotic Strain Clostridium butyricum TO-A Produces Butyrate by Utilizing Lactate and Acetate"

_ijms, 2025, doi:10.3390/ijms26072951_

Round 1

Reviewer 1 Report

Comments and Suggestions for Authors

This manuscript describes the production of butyrate, a helpful compound in the human intestine, by a strain of C. butyricum. The authors found that genes in the bacteria control relative amounts of D- and L-lactate, and that the inclusion of this strain in probiotics is warranted. The study appears to be well done, the procedures are appropriate, and the conclusions follow from the observations. Once published, it will advance our knowledge in the functioning of a butyrate-producing probiotic.

Lines 12-14: The "is unclear" part does not belong where it is. Perhaps "how it utilizes" would make it fit.

Line 40: Should be "bacteria".

Line 106: Define your abbreviation: ADL = acetate, D-lactose, L-lactose?

Line 136: AD = acetate D-lactose, AL = acetate L-lactose?

Reference 28: Use lower case letters in most of the title words.

Author Response

Our responses to Comments and Suggestions

Comments and Suggestions for Authors

This manuscript describes the production of butyrate, a helpful compound in the human intestine, by a strain of C. butyricum. The authors found that genes in the bacteria control relative amounts of D- and L-lactate, and that the inclusion of this strain in probiotics is warranted. The study appears to be well done, the procedures are appropriate, and the conclusions follow from the observations. Once published, it will advance our knowledge in the functioning of a butyrate-producing probiotic.

Thank you very much for reviewing our manuscript. We hope that our findings will provide scientific evidence for the mechanisms of action of clinically applied probiotics, as well as contribute to the evaluation methods for the development of novel butyrate-producing probiotics, particularly in treatment and prevention of bowel disease.

Comments 1: Lines 12-14: The "is unclear" part does not belong where it is. Perhaps "how it utilizes" would make it fit.

Response 1: We appreciate very much for the comment. The sentence is revised to: “Although CBTOA is known to increase intestinal butyrate levels, it is unclear how it utilizes lactate and acetate similar to LUB, to produce butyrate.” We hope that this revision clarifies the background and purpose of our research.

Comments 2: Line 40: Should be "bacteria".

Response 2: Thank you very much for the careful check. This is a spelling error. “bacterial” has been replaced with “bacteria”.

Comments 3: Line 106: Define your abbreviation: ADL = acetate, D-lactose, L-lactose?

Response 3: Thank you for your comment regarding the abbreviation “ADL” on line 106. We have edited the text in accordance with your suggestion. However, we would like to clarify that in the manuscript, “ADL” is an abbreviation for “acetate, D-lactate, and L-lactate”, but not an abbreviation for “acetate, D-lactose, and L- lactose”. We appreciate your careful review and have updated the manuscript accordingly.

Comments 4: Line 136: AD = acetate D-lactose, AL = acetate L-lactose?

Response 4: Thank you for pointing this out. Abbreviations for “AD” and “AL” on line 136. As clarified for line 106, in the manuscript, “AD” and “AL” are abbreviations for “Acetate, D-lactate” and “acetate, L-lactate” respectively. We have edited the text to ensure consistency throughout.

Comments 5: Reference 28: Use lower case letters in most of the title words.

Response 5: We appreciate very much for the comment. In accordance with the suggestion, we have revised the title to use lower case letters in most of the words.

Reviewer 2 Report

Comments and Suggestions for Authors

This study demonstrated that the probiotic strain Clostridium butyricum TO-A efficiently produced butyrate by utilizing exogenous lactate and acetate, potentially serving as a valuable butyrate supplier for gut health. The study was logically structured, and the methodology was clear. Here are my comments:

  1. The results presented in Figure 3 already exist in Table 2, and it is recommended that only Table 2 be retained.
  2. Tables 2 and 4, it is recommended that tables A and B be combined into one table so that the differences between the two groups can be easily compared.
  3. It is recommended that the authors provide some description of the experimental setup in the results section, e.g., what was added to PY+ADL medium, which would be helpful to the readers.
  4. Lines 192-194 “In this study, we report for the first time that CBTOA not only utilizes sugars but also lactate and acetate to produce butyrate”, from Table S1B, it could be found that in PY+A medium, CBTOA did not utilize acetate to produce butyrate, whereas in PY+ADL medium, the concentration of acetate was still positively increasing. So did this conclusion require more evidence?
  5. Line 195 “Therefore, CBTOA can utilize lac- 195 tate and acetate added to the medium instead of sugars to produce butyrate”, how to make sure CBTOA can utilize acetate?
  6. Line 208 “pH increased”, did the increase in pH mean the bacteria are starting to die?
  7. It would be better if RT-qPCR could be utilized to determine the expression of key genes.

Author Response

Our responses to Comments and Suggestions

Comments and Suggestions for Authors

This study demonstrated that the probiotic strain Clostridium butyricum TO-A efficiently produced butyrate by utilizing exogenous lactate and acetate, potentially serving as a valuable butyrate supplier for gut health. The study was logically structured, and the methodology was clear. Here are my comments:

Thank you very much for reviewing our manuscript. We hope that our findings will provide scientific evidence for the mechanisms of action of clinically applied probiotics, as well as contribute to the evaluation methods for the development of novel butyrate-producing probiotics, particularly in treatment and prevention of bowel disease.

Comments 1: The results presented in Figure 3 already exist in Table 2, and it is recommended that only Table 2 be retained.

Response 1: Thank you for your suggestion regarding the overlapping description between Figure 3 and Table 2. We agree that the information is sufficiently presented in Table 2. Therefore, we have removed Figure 3 as recommended and added the relevant p-value directly into the main text to express clarity and brevity of the results.

Comments 2: Tables 2 and 4, it is recommended that tables A and B be combined into one table so that the differences between the two groups can be easily compared.

Response 2: We appreciate very much for the comment. As you suggested, we have combined Tables 2A and 2B into Table 2, to make it easier to compare the data of two groups. We have also reorganized Table 4 in the same way.

Comments 3: It is recommended that the authors provide some description of the experimental setup in the results section, e.g., what was added to PY+ADL medium, which would be helpful to the readers.

Response 3: Thank you for your suggestion to briefly explain the experimental design in the Results section. To ensure clarity for readers, we have added text to the beginning of 2.2 and 2.3 with the interpretation of abbreviations to help readers understand the experimental flow.

Comments 4: Lines 192-194 “In this study, we report for the first time that CBTOA not only utilizes sugars but also lactate and acetate to produce butyrate”, from Table S1B, it could be found that in PY+A medium, CBTOA did not utilize acetate to produce butyrate, whereas in PY+ADL medium, the concentration of acetate was still positively increasing. So did this conclusion require more evidence?

Comments 5: Line 195 “Therefore, CBTOA can utilize lac-195tate and acetate added to the medium instead of sugars to produce butyrate”, how to make sure CBTOA can utilize acetate?

Response 4 and 5:

Thank you for your comments regarding the utilization of acetate by CBTOA. We understand that Comment 4 and 5 are both about the utilization of acetate, so we will address and respond to them together as follows.

In this study, CBTOA produced acetate in PY medium, which contained no added sugars or organic acids (Table 2). This indicates that CBTOA is capable of producing acetate. However, in PY+ADL and PY+AD media—which were supplemented with D-lactate and acetate—the net acetate production was lower than in PY medium. Moreover, the net acetate production in PY+AD, where higher amounts of butyrate were produced, was approximately equal to its initial concentration (Tables 2 and 4). These findings suggest a relationship between increased butyrate production and the consumption of acetate by CBTOA.

Additionally, to confirm whether CBTOA can utilize acetate alone, we cultured it in PY+A medium (supplemented only with acetate). Under these conditions, the acetate level remained comparable to that in PY medium, and no additional consumption of acetate or increase in butyrate production was observed. Therefore, in the absence of added lactate, CBTOA did not consume acetate, and butyrate production did not increase. Based on these results, we concluded that both exogenous lactate and acetate are required for CBTOA to effectively produce butyrate.

In the revised manuscript, we have added explanations about CBTOA ability to produce acetate and the net changes in acetate levels to clarify these points.

Comments 6: Line 208 “pH increased”, did the increase in pH mean the bacteria are starting to die?

Response 6: We appreciate very much for the comment regarding the relationship between the increase in pH and the viability of CBTOA. In this study, the increase in pH was observed in the PY+ADL and PY+AD media (Figures 2 and 3). In these media, the total bacterial count was higher than those in the PY+AL (Table 3) and PY (Table1) media, suggesting that the increase in pH was not related to CBTOA cell death. We believe that the pH increase was caused by the consumption of lactate. In fact, the lactate concentration in PY+ADL and PY+AD decreased compared to its initial concentration (Tables 2 and 4), and PY+AD exhibited a greater pH increase, which corresponds to its larger decrease in lactate.

Comments 7: It would be better if RT-qPCR could be utilized to determine the expression of key genes.

Response 7: Thank you very much for the valuable suggestion. Unfortunately, due to time constraints and limitations in our current research setup, we are unable to conduct these experiments for the present revision. However, we will certainly consider incorporating this approach in our future studies.